# Bifurcation Control on the Un-Linearizable Dynamic System via Washout Filters

**DOI:** 10.3390/s22239334

**Published:** 2022-11-30

**Authors:** Chi Zhai, Chunxi Yang, Jing Na

**Affiliations:** 1Faculty of Mechanical and Electrical Engineering, Kunming University of Science and Technology, Kunming 650093, China; 2Faculty of Chemical Engineering, Kunming University of Science and Technology, Kunming 650500, China

**Keywords:** Andronov–Hopf bifurcation, washout filter, normal-form degeneracy, feedback stabilization

## Abstract

Information fusion integrates aspects of data and knowledge mostly on the basis that system information is accumulative/distributive, but a subtle case emerges for a system with bifurcations, which is always un-linearizable and exacerbates information acquisition and presents a control problem. In this paper, the problem of an un-linearizable system related to system observation and control is addressed, and Andronov–Hopf bifurcation is taken as a typical example of an un-linearizable system and detailed. Firstly, the properties of a linear/linearized system is upon commented. Then, nonlinear degeneracy for the normal form of Andronov–Hopf bifurcation is analyzed, and it is deduced that the cubic terms are an integral part of the system. Afterwards, the theoretical study on feedback stabilization is conducted between the normal-form Andronov–Hopf bifurcation and its linearized counterpart, where stabilization using washout-filter-aided feedback is compared, and it is found that by synergistic controller design, the dual-conjugate-unstable eigenvalues can be stabilized by single stable washout filter. Finally, the high-dimensional ethanol fermentation model is taken as a case study to verify the proposed bifurcation control method.

## 1. Introduction

In nonlinear systems covering ecology, economics, sociology and many other fields, complex dynamic evolutionary behaviors, such as multiplicity, self-oscillation, and even chaos arise in the vicinity of bifurcation points [1], and various methods are adopted to enhance sensory ability for accurate manipulation and control: the bifurcation oscillator is applied as a quantum state amplifier [2], the oscillating clearance of the rolling-element bear is taken as the bifurcating parameter with repetitive errors fused into the control system [3], and deep learning is implemented for the control of the Van del Pol system [4]. Mathematically, a bifurcation takes place at a parameter value where the system loses structural stability with respect to parameter variations, i.e., a phase portrait around the equilibrium is not locally topologically conjugate to the phase portraits around the equilibrium at nearby parameter values [5,6], which leads the solution manifold to change significantly, and exacerbates system information acquisition and control implementation [7,8].

Bifurcation control [9] refers to the task of designing a controller to suppress or reduce existing bifurcating dynamics, thereby achieving desirable dynamical behaviors, i.e., to obtain an asymptotically stable phase portrait. Feedback is often applied on a system with bifurcations, where the methods of linear/nonlinear feedback, washout-filter-aided feedback, impulse control, vibrational control, delayed feedback [10], etc., are frequently adopted. Nonlinear analysis and control based on harmonic balance approximations or nonlinear frequency response is also studied [11]. Recently, the idea of the anti-control of bifurcations [12] is proposed to segregate unwanted dynamics by deliberately introducing simple bifurcations into the system.

However, bifurcation control sometimes exhibits contradictory outcomes: improper control actions or process–model mismatch might cause the closed-loop system to move across the stable manifold, leading to more complicated dynamic behaviors [13], and one cause is that many control laws are developed on the basis of linear system theory (gain-scheduling, feedback linearization, etc.), but most dynamic systems with bifurcation are un-linearizable, i.e., truncating out the nonlinear terms alters the structural stability of the original system, and the higher-ordered-terms might not shrink to zero as the time approaches infinity. On the other hand, normal form degeneracy provides an effective way to evaluate those nonlinear terms, where types of bifurcations are expressed with quadratic or cubic degeneracy, and state-feedback based on normal forms and invariant sets are developed [14,15].

The main goal of this paper was to address the problem of an un-linearizable system in bifurcation control, where a comparative study is conducted on both the original and linearized models using washout-filter-aided state feedback. Through washout filters, a feedback signal is expected to be well geared with the open-loop dynamics and generate asymptotic outcomes. Abed and co-workers [16,17,18,19] implemented washout filters for bifurcation control, and feedback stabilization theory is proposed through auxiliary state variables that are inter-coupled. For systems with varying bifurcation parameters, the adaptive control scheme is also developed [12]. Since washout-aided feedback is introduced to migrate out of unstable eigenvalues, Marquardt [20] suggested manipulating directly on the unstable eigenspace, and introducing a pair of complex conjugate eigenvalues instead. Zhang [21] further employed eigenvalue assignment to achieve bifurcation control with guaranteed transient dynamics. Chen [22] proposed a systematic way of choosing constants and control parameters for the control of a high-dimensional bifurcating system. However, for a complicated system, i.e., both single- and dual-unstable eigenvalues exist, and controller design could be rather sophisticated.

The aforementioned washout-filter-aided control was developed on the linearized system, and the linearization procedure might change the structural stability of the original system and cause redundant control actions. In this work, the elementary properties of a linearizable system are offered, which provides a comparison with the un-linearizable system when these properties fail. Then, the system under Andronov–Hopf bifurcation is taken as a typical case of an un-linearizable one and the related analysis and bifurcation control strategy are developed. Since washout-filter-aided control can migrate the unstable eigenvalues, numerical bifurcation analysis on the closed-loop system reveals that, by inserting only one washout filter to any of the dual-unstable eigenspaces, the system might be stabilized. Compared with previous developed control strategies [23], our method could downsize the numbers of washout filters, facilitating the application for more practical and general cases, especially for high-dimensional systems.

The inconsistency of feedback stabilization and numerical bifurcation analysis stems from the fact that many nonlinear systems with bifurcation are un-linearizable [24]. This work is organized as follows: Through the analysis of the normal form of Andronov–Hopf bifurcation, the properties of an un-linearizable system are detailed in this paper; then, washout-filter-aided feedback on the un-linearizable system is analyzed in contrast with previously developed washout-filter-aided control strategies; and the continuous ethanol fermentation model is taken as an example of the un-linearizable system and is discussed in detail, and finally, concluding remarks are provided last.

## 2. The Un-Linearizable System

### 2.1. Properties of the Linear System

Previously to the discussion on the nonlinear dynamic system, basic properties concerning the *linear system* are provided. Consider a system exhibiting outcomes (characters) *M*, and which is detected with finite numbers of sensible causes $x_{1},\ldots ,x_{n}$, related by function *f*; then, a linear/linearizable system starts as a weighted contribution of individual variables
(1)$$M=f\left(x_{1},x_{2},\cdots ,x_{n}\right)\approx \frac{\partial f}{\partial x_{1}}x_{1}+\frac{\partial f}{\partial x_{2}}x_{2}\cdots +\frac{\partial f}{\partial x_{n}}x_{n}.$$

Furthermore, the coupling relation between causes and outcomes is decoupled through variable substitution and combination, e.g., a 2 × 2 input–output system with a linearized form is given as follows,
(2)$$M=\left[\begin{array}{c} M_{1}\\ M_{2} \end{array}\right]=\left[\begin{array}{c} f_{1}\left(x_{1},x_{2}\right)\\ f_{1}\left(x_{1},x_{2}\right) \end{array}\right]\approx \left[\begin{array}{cc} \frac{\partial f_{1}}{\partial x_{1}} & \frac{\partial f_{1}}{\partial x_{2}}\\ \frac{\partial f_{2}}{\partial x_{1}} & \frac{\partial f_{2}}{\partial x_{2}} \end{array}\right]\left[\begin{array}{c} x_{1}\\ x_{2} \end{array}\right].$$

Assume that the Jacobian matrix is non-singular, and a coordinate change y=Ax exists that could diagonalize Equation (Equation 2)
(3)$$\begin{array}{cc} \; & \left[\begin{array}{c} M_{1}\\ M_{2} \end{array}\right]\approx \left[\begin{array}{cc} \frac{\partial f_{1}}{\partial x_{1}} & \frac{\partial f_{1}}{\partial x_{2}}\\ \frac{\partial f_{2}}{\partial x_{1}} & \frac{\partial f_{2}}{\partial x_{2}} \end{array}\right]\left[\begin{array}{c} x_{1}\\ x_{2} \end{array}\right]=A^{-1}\left[\begin{array}{cc} J_{11} & 0\\ 0 & J_{22} \end{array}\right]A\left[\begin{array}{c} x_{1}\\ x_{2} \end{array}\right]=A^{-1}\left[\begin{array}{c} J_{11}y_{1}\\ J_{22}y_{2} \end{array}\right],\\ \; & where:\\ \; & \left[\begin{array}{cc} \frac{\partial f_{1}}{\partial x_{1}} & \frac{\partial f_{1}}{\partial x_{2}}\\ \frac{\partial f_{2}}{\partial x_{1}} & \frac{\partial f_{2}}{\partial x_{2}} \end{array}\right]=A^{-1}\left[\begin{array}{cc} J_{11} & 0\\ 0 & J_{22} \end{array}\right]A,\;y=\left[\begin{array}{c} y_{1}\\ y_{2} \end{array}\right]=A\left[\begin{array}{c} x_{1}\\ x_{2} \end{array}\right]. \end{array}$$

Equation (Equation 3) indicates that for a linear/linearizable system, outcomes can be departed as a sum of the weighted causes separately, and sets of causes can be found to correlate every outcome exclusively.

The above linearsystem properties are held for the linear-time-invariant (LTI) dynamic system. Given in state-space form, a multi-input–multi-output (MIMO) system undergoes Laplace transform and obtains similar distributive and decoupling properties
(4)$$\left\{\begin{array}{c} x^{\prime }=Ax+Bu+\xi \left(x,u\right)\\ y=Cx+Du \end{array}\right.\overset{↔}{Laplace}\;\frac{y}{u}=G\left(s\right)\approx C\frac{B}{sI-A}+D,$$
where ξ is the nonlinear terms that are truncated out in the linearizable dynamic system, whilst *y* and *u* represent outcomes and causes, respectively.

Mention that many nonlinear controls are developed on the linearized system. In practical applications, the presence of system nonlinearities causes the dynamic behavior to be qualitatively different from one operating regime to another; hence, one might obtain linear (at a specific set-point) models valid within a “small” region about the linearization point, and perform local designs for a set of operating conditions and then construct a gain-scheduling scheme that interpolates controller gains as the process traverses the operating region.

Accordingly, un-linearizable dynamic systems are those truncating out ξ as it alters the structural stability of the original system, and many dynamic systems with bifurcation are un-linearizable. Since a system under Andronov–Hopf bifurcation has dual- unstable eigenvalues with planar phase portrait, which is a typical multivariate, un-linearizable case, the following analysis and control is focused on the systems with Andronov–Hopf bifurcation.

### 2.2. Normal Form Andronov–Hopf Bifurcation: A Typical Un-Linearizable System

A standard approach to study the behavior of ordinary differential equations (ODE) around a bifurcation point is through normal form analysis, which starts by center manifold degeneracy. For a parameterized dynamical system,
(5)$$x^{\prime }=f\left(x,a\right),$$
where $x\to f\left(x,\alpha \right)\in R^{n}$ and α is the bifurcation parameter, with equilibrium $x_{0}$ satisfying $f(x_{0},0)=0$. Assume eigenvalues of the Jacobian about $x_{0}$ are $\lambda_{1,2}=\pm i\omega ,\lambda_{k}<0\left(k=3,4,\ldots ,n\right)$, then, there exists n-2 invariant/center manifold that is tangent at the bifurcation point to the eigenspace of the neutrally stable eigenvalues $\lambda_{1,2}$. Specifically, for a small neighborhood of α∈(0,δ) and within the invariant manifold exists a 2-dimensional attracting manifold and the coordinate change of the first two eigenspace; by introducing complex conjugates *z* and $z^{*}$, Equation (Equation 5) around $x_{0}$ can be given as
(6)$$z^{\prime }=i\omega z+g\left(z,z^{*}\right)\;z\in C^{1},$$
where $g(z,z^{*})$ is the degenerated terms at α=0 and can be given explicitly through rules of implicit function theorem. Assume $S\left(z,z^{*}\right)\in R^{\left(n-2\right)}$ represents the stable manifold, the n-2 stable equivalence is provided as follows
(7)$$\omega \left(z^{*}\frac{\partial S}{\partial z}\left(z,z^{*}\right)-z\frac{\partial S}{\partial z^{*}}\left(z,z^{*}\right)\right)=f_{3,\ldots ,n}\left(S\left(z,z^{*}\right),p\left(z,z^{*}\right)\right),$$
and the n-2 solutions are exponentially convergent to S(t)
(8)$$x_{3,\ldots ,n}\left(t\right)=S\left(t\right)+O\left(e^{-\beta t}\right),ast\to \infty ,$$
where β is the positive convergent rate. Considering that $x_{3,...,n}$ are locally asymptotic and therefore can be neglected in a local stability analysis around the bifurcation point, in the following discussion, about the Andronov–Hopf bifurcation, a two-dimensional topologically equivalent eigenspace containing dual-unstable ($\lambda_{1,2}=\pm i\omega$) is discussed instead of Equation (Equation 5).

For a general two-dimensional parameterized system, i.e., $x=\left(x_{1},x_{2}\right)\to f\left(x,\alpha \right)\in R^{2},\;\alpha \in R$, linearization about the equilibrium gives
(9)$$\begin{array}{cc} \; & x^{\prime }=A\left(\alpha \right)x+h\left(x,\alpha \right),\\ \; & where:\\ \; & A\left(\alpha \right)=\left(\begin{array}{cc} a & b\\ c & d \end{array}\right), \end{array}$$
where α, *b*, *c* and *d* are elements of the Jacobian A(α), which is simplified as A;h(x,α) represents the higher-ordered term. The characteristic equation corresponding to *A* gives
(10)$$\begin{array}{cc} \; & \lambda^{2}-tr\left(A\right)\lambda +det\left(A\right)=0\\ \; & \Rightarrow \left\{\begin{array}{c} \lambda =\frac{1}{2}\left(tr\left(A\right)+\sqrt{tr{\left(A\right)}^{2}-4det\left(A\right)}\right)\\ \lambda^{*}=\frac{1}{2}\left(tr\left(A\right)-\sqrt{tr{\left(A\right)}^{2}-4det\left(A\right)}\right) \end{array}\right., \end{array}$$
where tr(A) and det(A) are the trace and determinant, respectively. Andronov–Hopf bifurcation takes place at α=0 satisfying tr(A)=0 and $det\left(A\right)=\omega_{0}^{2}$. Assume in a small neighborhood α∈(0,δ) that the eigenvalues are written as $\lambda_{1}\left(\alpha \right)=\mu +i\omega$ and $\lambda_{2}\left(\alpha \right)=\mu -i\omega$, where μ=tr(A) and $\omega^{2}=tr{\left(A\right)}^{2}-det\left(A\right)$; then, T(α) exists to transform A(α) into canonical real form as follows
(11)$$J=T\left(\alpha \right)A\left(\alpha \right)T^{-1}\left(\alpha \right)=\left(\begin{array}{cc} \mu (\alpha ) & -\omega (\alpha )\\ \omega (\alpha ) & \mu (\alpha ) \end{array}\right),$$
where $J^{-1}=J^{T}/\left(\mu^{2}+\omega^{2}\right)$. Let y=T(α)x and Equation (Equation 9) turns to
(12)$$\left\{\begin{array}{c} y_{1}^{\prime }=\mu \left(\alpha \right)y_{1}-\omega \left(\alpha \right)y_{2}+{o(|y|}^{2})\\ y_{2}^{\prime }=\omega \left(\alpha \right)y_{1}+\mu \left(\alpha \right)y_{2}+{o(|y|}^{2}) \end{array}\right..$$

Introducing $z=y_{1}+iy_{2}$ and $z^{*}=y_{1}-iy_{2}$ and Equation (Equation 12) presents one-dimensional equivalence as follows
(13)$$z^{\prime }=\lambda \left(\alpha \right)z+g\left(z,z^{*}\right),$$
with $g=o(z^{2})$.Let q(α) be the eigenvector of $\lambda_{1}$ about A(α) and one can give an eigenvector p(α) corresponding to $\lambda_{2}$. Since $\lambda_{1}$ and $\lambda_{2}$ are complex conjugates, *p* and *q* can be chosen so that the inner product satisfies 〈q,p〉=1, and
(14)$$\begin{array}{cc} \; & \langle p,q^{*}\rangle =\langle p,\frac{1}{\lambda_{2}}Aq^{*}\rangle =\frac{1}{\lambda_{2}}\langle A^{T}p,q^{*}\rangle =\frac{\lambda_{1}}{\lambda_{2}}\langle p,q^{*}\rangle \\ \; & \Rightarrow \langle p,q^{*}\rangle =0. \end{array}$$

Equation (Equation 11) after transposition has $T^{T}\left(\alpha \right)J^{T}=A^{T}\left(\alpha \right)T^{T}\left(\alpha \right)$, and if p(α) has the following form
(15)$$p\left(\alpha \right)=\left(\begin{array}{c} p_{1}\\ p_{2} \end{array}\right)=\left(\begin{array}{c} T_{11}-iT_{21}\\ T_{12}-iT_{22} \end{array}\right),$$
where $T_{ij}$ is *i*-th line, *j*-th row element, then *z* has the explicit form as follows
(16)z=〈p(α),x〉.

Considering 〈q,p〉=1 and Equation(13), the original variable *x* has the only equivalent transform
(17)$$x=\left(\begin{array}{c} q\\ q^{*} \end{array}\right)\left(z\;z^{*}\right).$$

Substituting Equation (Equation 17) into (16), one obtains
(18)$$z^{\prime }=\lambda \left(\alpha \right)z+\langle p\left(\alpha \right),F\left(zq+z^{*}q^{*},\alpha \right)\rangle .$$

Here, the nonlinear term F(x,α) in Equation (Equation 9) after Taylor expansion gives
(19)$$\begin{array}{cc} \; & F\left(x,0\right)=\frac{1}{2}B\left(x,x\right)+\frac{1}{6}C\left(x,x,x\right)+{O(||x||}^{4}),\\ \; & with\left\{\begin{array}{c} B_{1,2}\left(x,y\right)=\sum_{j,k=1}^{2}\frac{\partial^{2}F_{1,2}\left(\xi ,0\right)}{\partial \xi_{j}\partial \xi_{k}}{|}_{\xi =0}x_{j}y_{k}\\ C_{1,2}\left(x,y,m\right)=\sum_{j,k=1,l=1}^{2}\frac{\partial^{3}F_{1,2}\left(\xi ,0\right)}{\partial \xi_{j}\partial \xi_{k}\partial \xi_{m}}{|}_{\xi =0}x_{j}y_{k}m_{l} \end{array}\right.. \end{array}$$

Substituting Equation (Equation 19) into (18) and the second-order term B gives
(20)$$z^{\prime }=\lambda z+\frac{\langle p,B(q,q)\rangle }{2}z^{2}+\langle p,B\left(q,q^{*}\right)\rangle zz^{*}+\frac{\langle p,B\left(q^{*},q^{*}\right)\rangle }{2}z^{*2}+{O(|z|}^{3}).$$

Through a coordinate change in Equation (Equation 21) the second-order term vanishes and gives Equation (Equation 22)
(21)$$\begin{array}{cc} z=\omega +\frac{\langle p,B(q,q)\rangle }{2\lambda }\omega^{2}+ & \frac{\langle p,B\left(q,q^{*}\right)\rangle }{\lambda^{*}}\omega \omega^{*}+\frac{\langle p,B\left(q^{*},q^{*}\right)\rangle }{2(2\lambda^{*}-\lambda )}\omega^{*2} \end{array}$$
(22)$$\begin{array}{cc} \omega^{\prime } & =\lambda \omega +O(|\omega {|}^{3}) \end{array}$$

Similarly, the third-order term after coordinate change obtains
(23)$$\begin{array}{cc} \; & \omega^{\prime }=\lambda \omega +\sigma \omega^{2}\omega^{*}+{O(|\omega |}^{4}),\\ \; & where:\\ \; & \sigma =\frac{\langle p,C\left(q,q,q^{*}\right)\rangle }{2}. \end{array}$$

Next, let $\omega =\rho e^{i\phi }$ and Equation (Equation 23) turns to
(24)$$\begin{array}{cc} \; & \left\{\begin{array}{c} \rho^{\prime }=\rho \left(\alpha -\rho^{2}\right)+\Phi \left(\rho ,\varphi \right)\\ \varphi^{\prime }=1+\Psi \left(\rho ,\varphi \right) \end{array}\right.,\\ \; & where:\\ \; & \Phi ={O(|\rho |}^{4}{);\;\Psi =O(|\rho |}^{3}). \end{array}$$

Suppose that φ performs constant rate rotation and Equation (Equation 24) mapped to the rotating plane gives
(25)$$\begin{array}{cc} \; & \frac{d\rho }{d\varphi }=\frac{\rho (\alpha -\rho^{2})+\Phi }{1+\Psi }=\rho \left(\alpha -\rho^{2}\right)+\left(\rho ,\varphi \right),\\ \; & where:\\ \; & R=O(|\rho {|}^{4}). \end{array}$$

Assume that $\rho_{0}=0$ is the initial condition, solving Equation (Equation 24) that provides
(26)$$\rho =e^{\alpha \varphi }\rho_{0}+e^{\alpha \varphi }\frac{1-e^{2\alpha \varphi }}{2\alpha }\rho_{0}^{3}+O_{1}\left(\right|\rho_{0}{|}^{4}).$$

For a fixed α>0, there exists a limit cycle, and when φ=2π, the state $\rho_{1}$ would turn back to the initial point
(27)$$\rho_{1}=e^{2\pi \alpha }\rho_{0}-e^{2\pi \alpha }\left(2\pi +O\left(\alpha \right)\right)\rho_{0}^{3}+O_{2}\left(\right|\rho_{0}{|}^{4}).$$

Comparing Equations (26) and (27), it is found that the 4th- or higher-order terms are unaffected by the formation of limit cycles, and hence can be truncated out. Furthermore, an Andronov–Hopf bifurcation is structurally equivalent to the following standard form
(28)x^1′x^2′=α−11αx^1x^2±(x^12+x^22)x^1x^2.

As shown in the following schematic diagram, if Equation (Equation 28) is linearized by truncating out the nonlinear terms, the phase portrait shows that the outcome is asymptotically stable to the origin (Figure 1a); if the nonlinear terms at Equation (Equation 28) are preserved, the phase portraits of the Andronov–Hopf bifurcation would be super-critical (Figure 1b), where the periodic solution is stable, or sub-critical (Figure 1c), where the periodic solution is unstable, and the criterion for super- or sub-criticality is judged by the sign of the third-order term of Equation (Equation 28).

It is an obvious standard form of the Andronov–Hopf bifurcation system that differs from the linearized system as it either attracts or repels an oscillatory trajectory, while the linearized one exhibits a diverging trend in the whole phase portrait. Hence, the nonlinear terms in Equation (Equation 28) are indispensable, and a system with Andronov–Hopf bifurcation is un-linearizable.

Address that there exists two variables ${\hat{x}}_{1}$ and ${\hat{x}}_{2}$ interacting with one another to form a limit cycle around the Andronov–Hopf bifurcation point (α>0). Furthermore, Equation (Equation 28) can be represented one-dimensionally using complex variables, which means manipulating either ${\hat{x}}_{1}$ or ${\hat{x}}_{2}$ to affect the dynamic behavior of the system directly (or equally). Hence, introducing one state feedback suffices stabilizing a system with Andronov–Hopf bifurcation. Notion, the sign of the third-order term in Equation (Equation 28), decides whether the bifurcation is super-critical (−) or sub-critical (+), and one can refer to [21] for more details.

## 3. Stabilizing Normal Form Andronov–Hopf Bifurcation

### 3.1. Washout Filter as a State Manipulator

Since washout filters reject steady-state signals and passing transient ones, it could be used as a state manipulator and by fine-tuned feedback, eliminates the instabilities originating from open-loop bifurcations. Figure 2 is the washout-filter-aided controller design scheme: by inserting washout filters between states and outputs, transient dynamic behaviors are separated out that can be used to cancel out the perturbations of the system through feedback; while outputs preserve the steady state information, this is important for a control system, meaning washout filters are inoperative until vibrational signals are detected. Assume that $x_{k}$ is the *k*-th element of the state variables, and *d* is the filter time constant. The control law used in this paper is quite simple, and *v* is the reference input used to determine the actual operating point and *k* is a controller parameter.

### 3.2. Feedback Stabilization Based on the Linearized System

To stabilize the normal-form Andronov–Hopf bifurcation, washout-filter-aided control is constructed. Without loss of generality, the linearized system is extended as follows
(29)$$x^{\prime }=Ax+Bu+h\left(x,u\right),$$
where the control $u={\left[u_{1},u_{2}\right]}^{T}$ takes u=v+ky, and *k* is the feedback gain; *B* is an arbitrary input-state transfer matrix and h(x,u) represents the higher-ordered-terms.

Adding the auxiliary state satisfies dz/dt=y, and the closed-loop system is provided as follows
(30)$$\begin{array}{c} \left\{\begin{array}{c} x^{\prime }=Ax+Bu+h\left(x,u\right)\\ z^{\prime }=P\left(x-z\right)\\ u=K(x-z) \end{array}\right.. \end{array}$$

Stabilizing Equation (Equation 30) requires the following augmented matrix $A_{C}$ being Hurwitz
(31)$$\begin{array}{cc} \left(\begin{array}{c} x^{\prime }\\ z^{\prime } \end{array}\right) & =\left(\begin{array}{cc} A+BK & -BK\\ P & -P \end{array}\right)\left(\begin{array}{c} x\\ z \end{array}\right)\\ \; & =A_{C}\left(\begin{array}{c} x\\ z \end{array}\right). \end{array}$$

 **Theorem 1.** 
*If A is non-singular and (A,B) is stabilizable, there exists (P,k), which makes $A_{C}$ Hurwitz.*


**Proof.** $A_{C}$ after similarity transforms of $T_{1}$ and $T_{2}$ gives
(32)$$\begin{array}{cc} T_{1}A_{C}T_{1}^{-1} & =\left(\begin{array}{cc} I & 0\\ 0 & P^{-1} \end{array}\right)\left(\begin{array}{cc} A+BK & -BK\\ P & -P \end{array}\right){\left(\begin{array}{cc} I & 0\\ 0 & P^{-1} \end{array}\right)}^{-1}\\ \; & =\left(\begin{array}{cc} A+BK & -KBP\\ I & -P \end{array}\right),\\ T_{2}T_{1}A_{C}T_{1}^{-1}T_{2}^{-1} & =\left(\begin{array}{cc} I & M\\ 0 & I \end{array}\right)\left(\begin{array}{cc} A+BK & -BK\\ P & -P \end{array}\right){\left(\begin{array}{cc} I & M\\ 0 & I \end{array}\right)}^{-1}\\ \; & =\left(\begin{array}{cc} A+BK+M & -AM-KBM-M^{2}-KBP-MP\\ I & -M-P \end{array}\right). \end{array}$$Through the following small gain analysis, the control (P,K) is obtained
(33)$$\begin{array}{cc} \; & AM+KBM+M^{2}+KBP+MP=0,\\ \; & s.t.\left\{\begin{array}{c} P=\varepsilon P_{1}+O\left(\varepsilon^{2}\right)\\ M=M_{0}+\varepsilon M_{1}+O\left(\varepsilon^{2}\right) \end{array}\right.. \end{array}$$Canceling the first- and second-order perturbational terms provides
(34)$$\begin{array}{cc} \; & \left\{\begin{array}{c} \left(A+BK+M_{0}\right)M_{0}=0\\ \left(A+BK\right)M_{1}+AP_{1}=0 \end{array}\right.,\\ \; & where:\\ \; & M_{0}=-A-BK, \end{array}$$
which dictates,
(35)$$M_{1}=-AP_{1}{\left(A+BK\right)}^{-1}.$$Then, around $M_{0}$ in a small region, the only *M* satisfying Equation (Equation 33) is given
(36)$$\begin{array}{cc} M & =M_{0}+\varepsilon M_{1}+O\left(\varepsilon^{2}\right)\\ \; & =-A-BK-\varepsilon AP_{1}{\left(A+BK\right)}^{-1}+O\left(\varepsilon^{2}\right) \end{array}$$
with $A_{C}$ satisfying
(37)$$\begin{array}{cc} \; & T_{2}T_{1}A_{C}T_{1}^{-1}T_{2}^{-1}=\left(\begin{array}{cc} -\varepsilon AP_{1}{\left(A+BK\right)}^{-1}+O\left(\varepsilon^{2}\right) & 0\\ I & A_{C2}\left(2,2\right) \end{array}\right),\\ \; & where:\\ \; & A_{C2}\left(2,2\right)=A+BK+\varepsilon \left(AP_{1}{\left(A+BK\right)}^{-1}-P_{1}\right)+O\left(\varepsilon^{2}\right). \end{array}$$When (A,B) is stabilizable, A+BK can be designed Hurwitz. Since *A* is non-singular, setting $P_{1}=A^{-1}\left(A+BK\right)$, then $-AP_{1}{\left(A+BK\right)}^{-1}$ is Hurwitz, so as $A_{C2}$. Plus, proper feedback gain *k* can promise the good transient dynamics of the closed-loop system.One can design *P* based on above perturbation theory, but how many washout filters needed to stabilize the system is still a problem, the following lemma provides some insights. □

 **Lemma 1.** 
*The parity of the unstable eigenvalues of −P needs to be consistent with that of A to make $A_{C}$ in Equation (Equation 31) be Hurwitz.*


**Proof.** From Equation (31), one has det($A_{C}$) = det(*A*)det(−P); hence, sign(det(AC)) = sign(det(*A*)det(−P)). If all eigenvaluses of $A_{C}$ have a negative real part, then sign(det($A_{C}$)) = ${\left(-1\right)}^{2n}$, which leads to sign(det(*A*)det(−P)) = ${\left(-1\right)}^{n-l}{\left(-1\right)}^{n-r},$ where *l* represents the number of unstable eigenvalues of *A*, and *r* represents the corresponding numbers of −P, i.e., l+r is an even number.Therefore, to stabilize the normal form Andronov–Hopf bifurcation based on information of Equation (Equation 29), −P needs to be a 2 × 2 matrix. The Laplace transform between state and control gives
(38)$$\frac{u(s)}{x(s)}=K\frac{sI}{sI+P}.$$Obviously, when the state x(s) is one-dimensional, the washout filter has d=P, but for a multi-dimensional system, e.g., *P* is 2 × 2, and the parameters of the washout filters are difficult to provide
(39)$$\begin{array}{cc} \; & \frac{sI}{sI+\varepsilon p^{*}}=\frac{s}{\Delta }\left(\begin{array}{cc} s+\varepsilon p_{22}^{*} & -\varepsilon p_{12}^{*}\\ -\varepsilon p_{21}^{*} & s+\varepsilon p_{11}^{*} \end{array}\right),\\ \; & where:\\ \; & \Delta =s^{2}+\varepsilon \left(p_{11}^{*}+p_{22}^{*}\right)s+\varepsilon^{2}\left(p_{11}^{*}p_{22}^{*}-p_{12}^{*}p_{21}^{*}\right). \end{array}$$In accordance with the form of washout filters, the higher-ordered terms in Equation (Equation 39) need to be simplified. Considering only the transient dynamics are concerned with washout filters, and based on the Laplaceextremetheorem, *s* in Equation (Equation 39) takes maximal value, which leads to the idea of the form of washout filters
(40)$$\frac{sI}{sI+\varepsilon P^{*}}=\left(\begin{array}{cc} \frac{s}{s+\varepsilon (P_{11}^{*}+P_{22}^{*})} & 0\\ 0 & \frac{s}{s+\varepsilon (P_{11}^{*}+P_{22}^{*})} \end{array}\right).$$Meaning two washout filters with d=tr(P) aid the controller design, and Equation (40) is decoupled. □

### 3.3. Bifurcation Analysis on the Closed-Loop System

Consider only one washout filter is introduced to $x_{i}$ with feedback $u=K_{i}y$, and (i,j) pairing generates four possible cases for a two-input–two-output system. The normal form Andronov–Hopf bifurcation is augmented and h(x,u) explicitly presented
(41)$$\left\{\begin{array}{c} x^{\prime }=Ax+Bu+h\left(x,u\right)\\ z^{\prime }=x_{i}-dz=y \end{array}\right.\Rightarrow \left\{\begin{array}{c} x^{\prime }=\left(\begin{array}{cc} a & -1\\ 1 & a \end{array}\right)\left(\begin{array}{c} x_{1}\\ x_{2} \end{array}\right)-\left(x_{1}^{2}+x_{2}^{2}\right)\left(\begin{array}{c} x_{1}\\ x_{2} \end{array}\right)+K_{i}y\\ y^{\prime }=f_{i}\left(x_{i},K_{i}y\right)-dy \end{array}\right.,$$
where *y* is the washout-filter-output corresponding to $x_{i}$. Based on Routh–Hurwitz stability criterion [25], one can deduce the control gain $K_{i}$ of the washout filter that stabilizes Equation (Equation 41), and the Jacobean of the augmented system gives
(42)$$\tilde{A}=\left[\begin{array}{cc} \left(\begin{array}{cc} a & -1\\ 1 & a \end{array}\right)\; & {\left(BK\right)}_{i}\\ \frac{\partial f_{i}}{\partial x_{1}}\frac{\partial f_{i}}{\partial x_{2}} & {\left(BK\right)}_{i}-d \end{array}\right].$$

Expending Equation (Equation 42) along the last column gives the characteristic polynomial $P_{c}$
(43)$$\begin{array}{cc} P_{c} & =det(\tilde{A}-\lambda I_{2+1})\\ \; & =\left({\left(BK\right)}_{i}-d-\lambda \right)\left(\lambda_{-i}-\lambda \right)+{\left(\begin{array}{c} \frac{\partial f_{i}}{\partial x_{i}} \end{array}\right)}_{n}\cdot {\left(BK\right)}_{i}\left(\lambda_{-i}-\lambda \right)\\ \; & =\left[\left({\left(BK\right)}_{i}-d-\lambda \right)\left(\lambda -\lambda_{n}\right)-{\left(\frac{\partial f_{i}}{\partial x_{1}}\right)}_{n}\cdot {\left(BK\right)}_{i}\right]\left(\lambda_{-i}-\lambda \right). \end{array}$$

One can design pairs of ($K_{i},d$) to make eigenvalues of [∼] in Equation (Equation 43) stable, but $\lambda_{-i}$ is kept unchanged. Meaning that, for the linearized system, dual-unstable eigenvalues need two washout filters.

However, Equation (Equation 41) is a nonlinear control, and the dynamics of $f_{i}$ could be compensated through feedback. The Jacobian from Equation (42) ignores nonlinear correlations between system and control feedback, which leads to redundant control loops. When numerical bifurcation analysis is implemented, it is found that only one washout filter can stabilize the closed-loop system represented by Equation (Equation 41).

One can design controllers through the identification of critical points for the emergence/elimination of unstable eigenvalues. For the case of Equation (Equation 41), the limit cycles when α>0 need to be eliminated (or stabilized); hence, with the aid of a washout filter framed in Figure 2, Andronov–Hopf bifurcation points in the ($K_{i},d$) domain can belocated numerically (refer to Appendix A for more details).

Substantial configurations are provided in Figure 3, where the solid lines are Andronov–Hopf bifurcation curves. BT represents Bogdanov–Takens bifurcation, and ZH represents zero-Hopf bifurcation, both of which are codimension-2 bifurcations originating from Andronov–Hopf bifurcation curves. Since $x_{1}$ and $x_{2}$ in Equation (Equation 41) are somewhat symmetric, configurations **C1** and **C4** are identical, and so are **C2** and **C3**. For each stabilizable region, d>0 implies the analog signal after washout filters are stable, while d<0 means that the feedback signal is unstable.

As shown in Figure 3, the Andronov–Hopf bifurcation curves separate out parameter space $\left(K_{i},d\right)$ that stabilizes the closed-loop system. Mention d=0 in cases of **C1** and **C4** indicates that traditional PID control stabilizes the system. It is observed d<0 is able to stabilize the system from a nonlinear analysis perspective, but the unstable washout filter generates diverging signals which are dangerous for real-time applications where interference is unavoidable and is not recommended. Exclusively for d>0, it is found **C2** and **C3** have broader stable margins than **C1** and **C4**, which implies manipulating variables and control feedback could be chosen such that stabilizing control is easy to achieve.

Based on an analysis in Figure 3, numerical simulations on the Simulink platform were conducted, where α=0.1 for Equation (Equation 41). As is shown in Figure 4, when the parameter pair (−5, 0.5) is adopted for case **C1**, it is stabilizable. The perturbation output after washout filter damps down and so as the states $x_{1}$ and $x_{2}$, and a positive phase shift between y and $x_{1}$ implies that the oscillatory dynamics of the control feedback and forward signal are well geared to stabilize the closed-loop system. Since $x_{1}$ and $x_{2}$ are coupled in the self-oscillatory system, stabilizing $x_{1}$ would cause $x_{2}$ to damp down. Further study of the information acquisition by the washout filter shows that, the stable one obtains oscillatory signals, while the unstable one is exponentially divergent, where the overall stable character of the closed-loop system is obtained by the subtraction of the forward and backward signals. Hence, to stabilize the self-oscillatory dynamic system, stable washout filters are suggested.

## 4. Case Study

### 4.1. Bifurcation Analysis on the Continuous Ethanol Fermentation Model

One of the most important bio-industrial sectors is the production of ethanol, which can be used as a gasoline additive to improve vehicle performance and reduce carbon emission. However, complex microorganism growth kinetics and ethanol inhibition present challenges for continuous production at the industrial scale, which limits the contribution of bio-fuel to current energy supply, and hence an exquisite controller design is critical to stabilizing the process. The structured model for Zymomonasmobilis fermentation [26] in a continuous stirred reactor is provided as follows, which considers the mass balance of the substrate ($C_{s}$, g/L), microorganism ($C_{x}$, g/L), *K*-compartment biomass ($C_{e}$, g/L), product ($C_{p}$, g/L) and the related parameters are given in Table 1
(44)$$\begin{array}{cc} \; & \frac{dC_{s}}{d_{t}}=\frac{\mu \cdot C_{S}\cdot C_{e}}{Y_{sx}\cdot \left(K_{S}+C_{S}\right)}-m_{S}\cdot C_{x}+D\cdot \left(C_{sf}-C_{s}\right)\\ \; & \frac{dC_{X}}{dt}=\frac{\mu_{max}\cdot C_{s}\cdot C_{e}}{K_{s}+C_{s}}-DC_{x}\\ \; & \frac{dC_{e}}{d_{t}}=\frac{\left(k_{1}-k_{2}C_{p}+k_{3}C_{P}^{2}\right)C_{s}\cdot C_{e}}{\left(K_{s}+C_{s}\right)}-DC_{e}\\ \; & \frac{dC_{p}}{d_{t}}=\frac{\mu_{max}\cdot C_{s}\cdot C_{e}}{Y_{px}\cdot \left(K_{s}+C_{s}\right)}+m_{p}\cdot C_{x}-DC_{p} \end{array},$$
where the dilution rate *D* and the substrate feed concentration $C_{sf}$ are set as the design /operation parameters because of practical concerns. *D* is the reciprocal of the residence time for the reaction stream, which has a fundamental impact on the bioreactor behavior; $C_{sf}$ always comes from upstream, which may be interfered with due to upstream uncertainties.

With this model, nonlinear analysis is conducted on the open-loop system. A two-parameter bifurcation diagram decomposes the parameter space ($D,C_{sf}$) into different sections, and each has distinct dynamic properties. Furthermore, as is shown in Figure 5, the Andronov–Hopf bifurcation curve divides the parameter space into S1 and S2. Any designing point at S1 generates a self-sustained oscillator, while the point at S2 is in stable steady equilibrium. Particularly, when $C_{sf}$ is set as 140 g/L, **H** is detected at $D_{H}=0.05024$ h${}^{-1}$. **H** is a supercritical Hopf bifurcation point due to its negative first Lyapunov coefficient (lp), and operating the process with *D* at the left hand side of **H** leads to self-oscillations, while stable equilibriums are obtained on the right-hand side. It is obvious that the highest $C_{p}$ is achieved when *D* is infinitesimal, which means that the reactor size needs to be very large. Since a high $C_{p}$ of the bioreactor outlet cuts down the separation cost of later processing, one should design the reactor by setting *D* as the small reasonable equipment cost.

We set the design point for Equation (Equation 44) as $P_{d}\left(0.022,140\right)$ in this work, which falls into the self-oscillatory section. The idea of designing a self-oscillatory process was inspired by [27,28]. Taking $P_{d}$ point as example, time average $C_{p}$ is 8.8% higher than the corresponding steady-state output of the same design condition. However, the presence of self-oscillatory dynamics brings a challenge for the control of the continuous system; therefore, bifurcation control with the aid of washout filters is demanded to stabilize this multi-dimensional system.

### 4.2. Washout-Filter-Aided Control of the Fermentation Model

The model in Equation (Equation 44) is four-dimensional, and dual-unstable eigenvalues exist when Andronov–Hopf bifurcations emerge. For $P_{d}$, the equilibrium for $C_{s},C_{x},C_{e}$ and $C_{p}$ are 0.8509 g/L, 21.6547 g/L, 0.01915 g/L and 58.71 g/L, respectively; and the corresponding eigenvalues are −2,−0.07136,0.01232±0.1411. Dual-unstable eigenspace emerges between states $C_{e}$ and $C_{p}$, indicating that the controlling variable would be chosen among them, $C_{p}$ is set as the washout filter input, and it is assumed that $C_{p}$ can be detected with online measuring devices. Then, the washout filter output is provided
(45)$$\frac{y}{C_{p}}=\frac{s}{s+d}\Rightarrow y^{\prime }=\frac{\mu_{max}\cdot C_{s}\cdot C_{e}}{Y_{px}\cdot \left(K_{s}+C_{s}\right)}+m_{p}\cdot C_{x}-DC_{p}-dy,$$
where *d* is the filter parameter, and to promise that *y* is stable and convergent, *d* needs to be positive. The manipulating variable is set to *D*, which is related to the input flow rate. Compared to the nominal $D_{0}$ = 0.022 h${}^{-1}$, the feedback by *P* control is provided
(46)$$D=D_{0}+ky,$$
where *k* is the feedback gain.

Combining Equations (44)–(46), one could implement bifurcation analysis on the closed-loop system. As is shown in Figure 6, when the washout-filter-aided P-control is facilitated on the continuous fermentation model, the controlling space (k,d) is separated into three sections by the Andronov–Hopf and neutral saddle curves, where only the upper-left one is stabilizable. Note that both Andronov–Hopf and neutral saddle bifurcations are codimension-1 bifurcations, and the bifurcations happening on these two curves are codimension-2 bifurcations. For the Andronov–Hopf bifurcation curve, three codimension-2 bifurcations, GH1 (0.1745, 0.0024), GH2 (0.0623, 0.0006) and ZH (0, 0.0005) are detected; and for the neutral saddle bifurcation curve, HH (0.02221, −0.03037) is detected. Mention for a given filter parameter, e.g., d=2, with the decrease in *k*, that the dynamics of the closed-loop system evolves from stable to saddle, and then diverges.

Further study is conducted by setting the control parameters to constant, e.g., d=2 and k=1, and observing the changes in the dynamics. As shown in Figure 7, between the two Andronov–Hopf bifurcation curves is the segment that can be stabilized by washout-filter-aided P-control. When the control is facilitated, stabilizing the designing point demands that $C_{p}$ reaches the steady state, and then the washout filter output *y* has a tendency to converge towards zero, and Equation (Equation 45) tends towards zero; hence, the closed-loop system has the same equilibrium as the open-loop one, and $C_{p}=58.71$ g/L is preserved at operational point $P_{d}$. As provided in Figure 8, the simulation results show converged dynamics after control d=2 and k=1, where $C_{p}$ and *D* reach $P_{d}$ as time approaches infinity.

## 5. Conclusions

In the presented study, we have focused on a subtle case with bifurcations, which is always un-linearizable and exacerbates information acquisition and a control problem. Taking the system with Andronov–Hopf bifurcation as a typical example, normal form analysis reveals that the cubic terms are the integral part of the whole system, and with the consideration of the un-linearizable characteristics, the stabilization procedure adopting washout-filter-aided control could realize the migration of dual-unstable eigenvalues with only one washout filter, which is not possible by previous control strategies. Moreover, the continuous ethanol fermentation process is taken as a case study for stabilizing a high-dimensional Andronov–Hopf bifurcation system: (1) variables mapping to the dual-unstable eigenspace are chosen as the manipulated variable; and (2) a stable washout filter parameter is demanded to stabilize the dual-unstable eigenvalues. By washout-filter-aided P-control, the ethanol fermentation process is stabilized and the equilibrium is preserved.

## Figures and Tables

**Figure 1 sensors-22-09334-f001:**
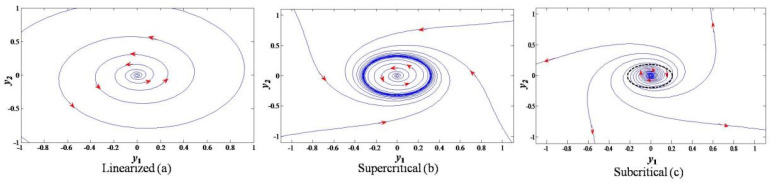
Phase portrait of (**a**) linearized system, (**b**) supercritical; and (**c**) subcritical Andronov–Hopf bifurcations, where α=0.1, and the arrows point out the time evolution direction.

**Figure 2 sensors-22-09334-f002:**
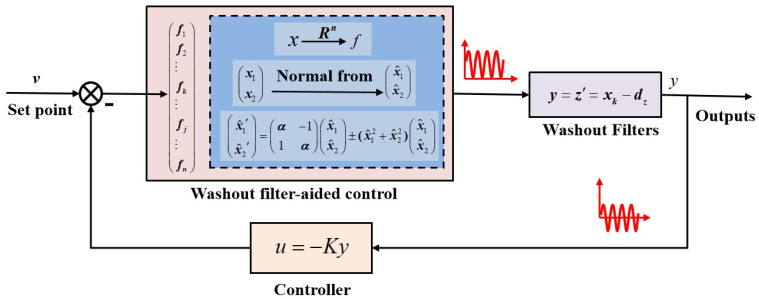
Block diagram for bifurcation control with washout-filter-aided stabilization.

**Figure 3 sensors-22-09334-f003:**
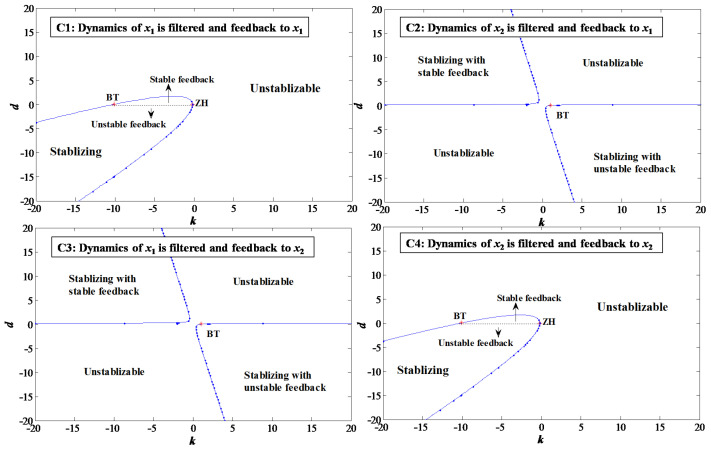
Analysis on the closed system of normal form Andronov–Hopf bifurcation.

**Figure 4 sensors-22-09334-f004:**
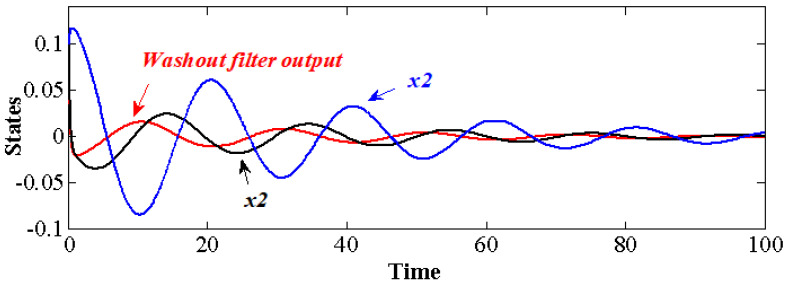
Time series of the closed-loop system, where the washout-filter-aided feedback parameters are d=0.5,k=−5.

**Figure 5 sensors-22-09334-f005:**
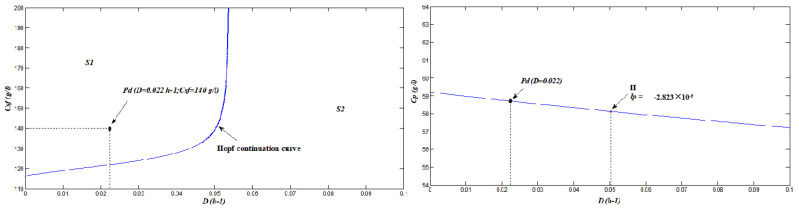
Bifurcation analyses on the ethanol fermentation process: two-parameter continuation diagram (**left**); and one-parameter continuation diagram (**right**).

**Figure 6 sensors-22-09334-f006:**
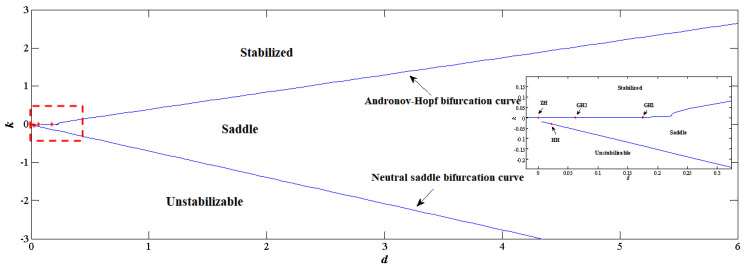
Two–parameter continuation diagram for the control variables, where **GH1** and **GH1** are the general Hopf bifurcation, **ZH** is the zero Hopf bifurcation and **HH** is the Hopf–Hopf bifurcation.

**Figure 7 sensors-22-09334-f007:**
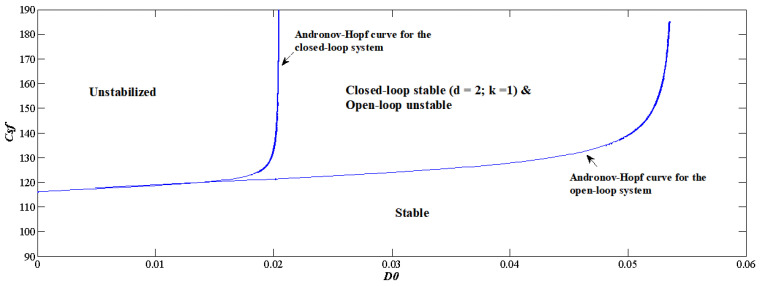
Two-parameter continuation diagram for the designing parameters.

**Figure 8 sensors-22-09334-f008:**
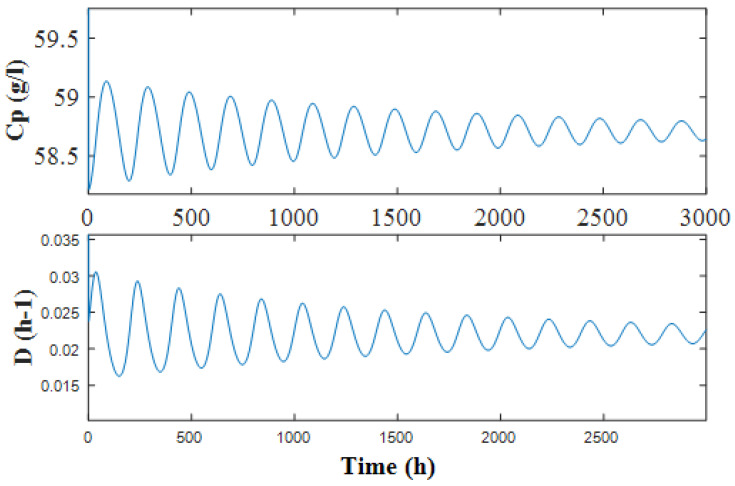
Stabilized outcomes for the closed-loop system, where d=2 and k=1.

**Table 1 sensors-22-09334-t001:** Parameters used for the continuous bio-ethanol reactor model.

Parameter	Definition	Value
$\mu_{max}$	Maximum specific biomass growth rate (h${}^{-1}$)	1.0
$Y_{sx}$	Yield factor for substrate to biomass (g/g)	0.00244498
$Y_{px}$	Yield factor for substrate to product (g/g)	0.00526315
$m_{s}$	Maintenance substrate consumption rate (g/g h${}^{-1}$)	2.16
$K_{s}$	Monod constant (g/L)	0.5
$k_{1}$	Empirical constant (h${}^{-1}$)	16
$k_{2}$	Empirical constant (L/g h${}^{-1}$)	0.497
$k_{3}$	Empirical constant (L${}^{2}$/g${}^{2}$h${}^{-1}$)	0.000383
$m_{p}$	Maintenance product consumption rate (g/g h${}^{-1}$)	1.1
$C_{sf}$	Substrate feed concentration (g/L)	140

## Data Availability

All data generated or analyzed during this study are included in this published article.

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
