# Peer review of "Bifurcation Control on the Un-Linearizable Dynamic System via Washout Filters"

_sensors, 2022, doi:10.3390/s22239334_

Round 1

Reviewer 1 Report

There are some problems should be addressed as mentioned in the attachment.

Author Response

Dear Reviewer,

We appreciate the chance to revise the paper, and also thank the reviewer for the constructive remarks.  The following is also a point-to-point response to the reviewer’s comments.

Best regards,

Chi Zhai, Ph. D

Faculty of Chemical Engineering

Kunming University of Science and Technology

Chemical Engineering Building, Room 314

727 Jingming South Road, Kunming, China, 650500

Office: +86 136 9328 5751

Reviewer 2 Report

In this paper, the problem of un-linearizable system related to system observation and control is addressed, and Andronov-Hopf bifurcation is taken as a typical example of un-linearizable system and detailed.

Comments:

1. The actual meaning of variables in the model needs to be explained. Such as Cs, Cx,Ce,Cp in model (44).

2. I suggest the author explain the selection of parameters in numerical simulations. I guess they may have biological significance.

3. The author reviewed some work on branches, but there are some new work suggestions. For example: Spatiotemporal dynamics induced by nonlocal competition in a diffusive predator-prey system with habitat complexity. Nonlinear Dynamics, 110, 879–900 (2022). https://doi.org/10.1007/s11071-022-07625-x; 

4.The author is suggested to improve the language of the manuscript. In Abstract: “the normal form Andronov-Hopf bifurcation” should be “the normal form of Andronov-Hopf bifurcation” The last sentence in the Abstract is too long. It is suggested to modify it.

Author Response

Dear Reviewer,

We appreciate the chance to revise the paper, and also thank the reviewer for the constructive remarks. Hereby we submit a revised version of our manuscript entitled “Study on the nonlinear dynamics of the continuous stirred tank reactors” (ID: Processes-959015). The following is also a point-to-point response to the reviewer’s comments.

Best regards,

Chi Zhai, Ph. D

Faculty of Chemical Engineering

Kunming University of Science and Technology

Chemical Engineering Building, Room 314

727 Jingming South Road, Kunming, China, 650500

Office: +86 136 9328 5751

Author Response

Dear Reviewer,

We appreciate the chance to revise the paper, and also thank the reviewer for the constructive remarks.The following is also a point-to-point response to the reviewer’s comments.

Best regards,

Chi Zhai, Ph. D

Faculty of Chemical Engineering

Kunming University of Science and Technology

Chemical Engineering Building, Room 314

727 Jingming South Road, Kunming, China, 650500

Office: +86 136 9328 5751
